# Impact of Lifestyle Modifications on Alterations in Lipid and Glycemic Profiles and Uric Acid Values in a Pediatric Population

**DOI:** 10.3390/nu14051034

**Published:** 2022-02-28

**Authors:** Marco Giussani, Antonina Orlando, Elena Tassistro, Giulia Lieti, Ilenia Patti, Laura Antolini, Gianfranco Parati, Simonetta Genovesi

**Affiliations:** 1Cardiologic Unit, IRCCS, Istituto Auxologico Italiano, 20100 Milan, Italy; dottormarcogiussani@gmail.com (M.G.); a.orlando@auxologico.it (A.O.); gianfranco.parati@unimib.it (G.P.); 2School of Medicine and Surgery, University of Milano—Bicocca, 20900 Monza, Italy; e.tassistro@campus.unimib.it (E.T.); g.lieti@campus.unimib.it (G.L.); a.patti8@campus.unimib.it (I.P.); laura.antolini@unimib.it (L.A.)

**Keywords:** children, lifestyle modifications, cardiovascular risk, overweight, obesity, dyslipidemia, insulin resistance, uric acid

## Abstract

Cardiometabolic risk factors are frequent in children and adolescents with excess weight. The aim of this study was to evaluate the effects of lifestyle modifications on alterations in lipid and glycemic profiles and uric acid values in a pediatric population at increased cardiovascular risk. The study involved 276 subjects with a mean age of 10.6 (2.3) years. Body mass index (BMI) z-score and biochemical parameters (serum low-density lipoprotein (LDL) cholesterol, triglycerides and uric acid and homeostasis model assessment to quantify insulin resistance (HOMA index)) were assessed at baseline and at the end of a median follow-up of 14.7 (12.4, 19.3) months. Throughout follow-up, all children received a non-pharmacological treatment based on increased physical activity, reduced sedentary activity and administration of a personalized, healthy and balanced diet. All children attended periodic quarterly control visits during follow-up. Multivariable statistical analyses showed that each BMI z-score point reduction at follow-up was associated with an 8.9 (95% CI −14.2; −3.6) mg/dL decrease in LDL cholesterol (*p* = 0.001), 20.4 (95% CI −30.0; −10.7) mg/dL in triglycerides (*p* < 0.001), 1.6 (95% CI −2.2; −1.0) in HOMA index (*p* < 0.001), and 0.42 (95% CI −0.66; −0.18) mg/dL in uric acid (*p* = 0.001) values. At each reduction of the BMI z-score by one point, the odds of presenting with insulin resistance and hyperuricemia at follow-up significantly decreased (OR 0.23, 95% CI 0.10–0.50, and OR 0.32, 95% CI 0.10–0.95, *p* < 0.001 and *p* < 0.05, respectively). Improvement of dietary habits and lifestyles may improve lipid and glycemic profiles and serum uric acid values in a pediatric population.

## 1. Introduction

Healthy lifestyles and diets are recommended for everyone and at all ages, to maintain good health and prevent non-communicable diseases. In the presence of cardiovascular risk factors, all guidelines propose a dietary-behavioral intervention as a first step [1,2,3,4]. This approach is especially recommended for children and adolescents, populations in which medication is used only in very select cases [5,6,7]. Although in children there is no precise definition of metabolic syndrome due to the lack of agreement of scientific societies on the cut off values of individual parameters, it is accepted that, even in childhood and adolescence, obesity is frequently associated with excess visceral adiposity, glucose metabolism disorders, increased triglycerides and decreased HDL cholesterol. In addition, an increase in serum uric acid values has been shown in overweight and/or hypertensive children and adolescents [8,9,10]. Therefore, cardio-metabolic risk factors, such alterations in lipid and glycemic profile and hyperuricemia, are frequently present in children and adolescents that are overweight or obese. There is evidence on the effectiveness of nonpharmacological dietary-behavioral treatment in reducing both excess weight and blood pressure values in children and adolescents with excess weight [11,12]. Two meta-analyses showed a positive effect of lifestyle modification and exercise training on LDL cholesterol and triglyceride values and insulin resistance markers in this population [13,14]. However, in the scientific literature, the information on the effectiveness of dietary-behavioral intervention administered as a therapy in children and adolescents with excess weight to correct altered metabolic profiles is still scarce and fragmentary. The aim of this study is to evaluate the effects of dietary-behavioral treatment on alterations in lipid and glycemic profiles and uric acid values in a population of children and adolescents referred to a second-level outpatient clinic for cardiovascular risk assessment in pediatric age.

## 2. Materials and Methods

### 2.1. Subjects

We studied a cohort of 276 children and adolescents (4–18 years), consecutively referred from 16 December 2002 to 12 September 2017 to our Unit for Cardiovascular Risk Assessment in Children (at the Istituto Auxologico Italiano, Milan, Italy) by their primary care pediatricians, because of evidence of excess weight and/or alterations in the lipid and/or glycemic profile and/or plasma uric acid values. Exclusion criteria were: impaired glucose tolerance, diabetes, any form of secondary hypertension, and treatment with antihypertensive drugs. The Unit for Cardiovascular Risk Assessment in Children included a pediatrician, a cardiologist, a nephrologist and a nutrition expert. The team interacted to cover all clinical aspects related to the patients.

### 2.2. Baseline and Follow-Up Assessments

In all children, anthropometric and biochemical parameters were assessed once at baseline and a second time at the end of follow-up. Between the baseline and the final follow-up assessment, periodic visits were performed every three months, during which the children’s anthropometric parameters were again recorded. The median follow-up was 14.7 (12.4, 19.3) months. Between the baseline and final assessments, a minimum of three visits and a maximum of six visits were performed in each child. All children consecutively referred to our center who had two complete assessments (at baseline and follow-up) of all outcomes (anthropometric parameters, complete lipid profile, glycemic profile with insulin dosage, and uric acid) were considered for analysis (Appendix A). Follow-up ended in September 2017.

### 2.3. Anthropometric Parameters

Height, weight and waist circumference (WC) were measured. Weight was approximated to the nearest 100 g, and height to the nearest 1 mm. Waist circumference was measured to the nearest 0.5 cm by a non-elastic flexible tape in standing position. Body mass index (BMI) was calculated as weight (kg)/height (m)^2^. Waist-to-height-ratio (WtHr) was calculated by dividing WC by height. BMI z-scores were calculated using the Centre for Disease and Control prevention charts available at https://www.cdc.gov/growthcharts/clinical_charts.htm (accessed on 27 January 2022). Weight class was defined according to the International Obesity Task Force classification [15], distinguishing between normal weight (NW), overweight (OW) and obese (OB) classes. Pubertal stage was assessed by a medical examination and children were classified into two categories: pre-pubertal and pubertal according to Tanner [16], considering prepubertal boys with gonadal stage 1 and girls with breast stage 1.

### 2.4. Biochemical Parameters

Blood samples were taken from all subjects after a 12-h fasting period in order to measure serum concentrations of total cholesterol, high density lipoprotein (HDL), triglycerides, glucose, insulin, uric acid and creatinine. Commercial kits, normally used for routine examinations of patients, were employed for all analyses. In detail: enzymatic colorimetric test Cholesterol Gen.2 Cobas Roche, for total cholesterol assay; colorimetric enzymatic test in homogeneous phase HDL-Cholesterol Gen.4 Cobas Roche, for HDL cholesterol; enzymatic colorimetric test Triglycerides Cobas Roche, for triglycerides assay; enzymatic method with hexokinase Glucose HK Gen.3 Cobas Roche, for glucose assay; immunoassay in ElectroChemiLuminescence Elecsys Insulin Cobas Roche, for insulin assay; colorimetric enzymatic test Uric Acid 2 Cobas Roche, for uric acid assay; and the colorimetric kinetic test based on the Jaffé method, Creatinine Jaffé Gen.2 Cobas Roche, for creatinine assay. LDL cholesterol was calculated using Friedewald’s formula, LDL cholesterol = total cholesterol − [HDL cholesterol + (triglyceridemia/5)]. HOMA index was calculated by dividing the product of serum insulin (µU/mL) and serum glucose (mmol/L) by 22.5 [17]. Glomerular filtration rate was estimated (eGFR) by means of the Schwartz formula using serum creatinine and height measurements and a k constant of 0.55 [18].

### 2.5. Definition of Metabolic Variables

The definition of dyslipidemia was based on the reference values of the National Cholesterol Education Program (NCEP) Expert Panel on Cholesterol Levels in Children: LDL cholesterol ≥ 130 mg/dL, HDL cholesterol < 40 mg/dL, triglycerides ≥ 100 mg/dL (children < 10 years), or ≥130 mg/dL (children ≥ 10 years) [19]. Dyslipidemia was defined as the presence of altered values for at least one of these parameters (LDL cholesterol, HDL cholesterol, triglycerides).

Insulin resistance was defined as the presence of HOMA Index values > 90th percentile according to sex and age in normal-weight children and adolescents [20].

Hyperuricemia was defined as the presence of uric acid values greater than the 90th percentile for sex and age according to [21] 90th percentile cut-off points for boys, 4.5 mg/dL for 6–9 years, 5.7 mg/dL for 10–13 years, and 6.4 for 14–17 years; for girls, 4.8 mg/dL for 6–9 years, 5.2 mg/dL for 10–13 years, and 5.3 for 14–17 years.

### 2.6. Recommended Lifestyle Modifications

All the children and adolescents were advised to perform at least two/three hours of structured physical activity every week [22], to increase the time dedicated to movement-play and to reduce sedentary activities (videogames or TV watching) to no more than one hour daily, following the recommendations of the Italian Society of Pediatrics (https://sip.it/2017/09/25/la-sip-presenta-la-piramide-dellattivita-motoria/) (accessed on 28 January 2022).

The participants received general advice for a healthy and balanced diet (increasing the consumption of fruit, vegetables, non-fat milk and dairy products, and reducing the intake of simple sugars and eliminate soft drinks) with adequate salt consumption (no more than 5 g/day, i.e., 2 g of sodium) according to the World Health Organization (WHO) guidelines (www.who.int/nutrition/publications/guidelines/sodium_intake/en/) (accessed on 27 January 2022).

During the baseline visit, an expert nutritionist assessed the physical activity and eating habits of the children by interviewing their parents/caregivers. Based on the obtained data, appropriate changes in lifestyle and nutrition were proposed. A nutritional analysis was performed in order to prepare a personalized dietary scheme for each child, by a dedicated software (Dietosystem, DS Medica S.r.l., Milan, Italy). Balanced dietary patterns for what regards the content of macronutrients (carbohydrates 55%, lipids 30%, proteins 15% of energy intake) and correct content of free sugars, saturated fatty acids and fiber were then proposed to all participants.

Depending on the pathological conditions (excess weight, alteration of lipid profile, alteration of glycemic profile, elevated uric acid values), specific recommendations were provided. When a child presented more than one of these conditions, a combination of the different dietary patterns was proposed. For each pathological condition, the rationale for the dietary intervention was explained to caregivers, to stimulate caregiver and family empowerment.

*Excess weight* OW or OB subjects were supplied with a weekly dietary scheme, whose caloric content had been calculated on the basis of both the basal metabolic rate by the Schofield equation [23] and the functional metabolism [24], through the analysis of the child’s physical activity and the usual energy expenditure during an average day. In younger children, a balanced dietary model was proposed, equal to the calculated energy expenditure (normocaloric regime), whereas in adolescents with severe excess weight, a mildly hypocaloric regime (−10%) was suggested.

*Alterations of lipid profile* Children with hypercholesterolemia were advised to completely eliminate foods containing trans-fatty acids, to reduce energy intake from saturated fatty acids to less than 5–10%, and to limit consumption of some meats, dairy products, chocolate, baked goods and fried and processed foods. Instead, the consumption of foods containing mono- and polyunsaturated fatty acids was favored. The intake of dietary cholesterol was not allowed to exceed 100 mg per 1000 kcal of the diet; to this end, the intake of foods of animal origin such as meat, egg yolks, shellfish and whole milk dairy products was limited, whereas the consumption of foods containing soluble fiber (whole grains, vegetables and legumes) and phytosterols was favored. In case of increased triglyceride levels at baseline, dietary advice included, besides aiming at a reduction of body weight, a limitation of calories deriving from free sugars to less than 10% of total calories, and in particular a reduction in the intake of fructose through the complete abolition of sugary drinks. In the case of mixed dyslipidemia, all the above indications were recommended and, in all participants, the consumption of fish (salmon, tuna and blue fish) was encouraged to increase the intake of ω3 [4,25].

*Insulin resistance* The dietary-behavioral treatment implemented to reduce HOMA index values coincided with that to reduce excess weight (see above). From a qualitative point of view, the consumption of non-starchy vegetables and fruits rich in fiber, vitamins, and minerals citrus fruits, legumes and preferably whole grains, lean meats, fresh cheeses, fish and nuts, and unsweetened dairy products was encouraged. On the other hand, the intake of sugary drinks, fruit juices, carbonated and soft drinks, starchy vegetables, such as potatoes, pumpkin and corn, processed snacks and canned foods sugary sweets, ice cream and chocolate was limited.

*Elevated uric acid* Children with high uric acid levels were instructed to completely eliminate all sugary drinks from their diet (soft and energy drinks, ready-to-drink teas, fruit juices and nectars, etc.) and to reduce consumption of sweet foods. To caregivers, a list of foods with high purine content was provided, with the recommendation to reduce as much as possible the intake of these foods (red meat, lamb and pork, sweetbreads, seafood and especially shellfish such as shrimps, lobsters, mussels, anchovies and sardines). The recommended foods with low purine content comprised non-fat dairy products, fresh fruit and vegetables, cereals and derivatives. Eggs, fish, chicken and white meats were allowed in moderation.

At each periodic visit, information to assess the compliance with diet modifications, increase in physical activity, decrease in time spent watching television or with video-games was requested from family members according to a panel of questions, and the answers were reported in the medical record. Reinforcement of dietary and of lifestyle recommendations was given when necessary.

### 2.7. Statistical Analysis

The characteristics of the cohort, overall, at baseline and at follow-up, were described as mean and standard deviation (SD) or median and interquartile range (IQR) if the variables were continuous, and as frequencies and percentages if they were categorical. Univariate analyses to compare the characteristics of the children at baseline and at follow-up were conducted through the *t*-test for paired data or the Wilcoxon signed-rank test for continuous variables, and by the McNemar test for categorical variables.

The Z-score value was calculated by the equation ((BMI value/M)^L^ − 1)/(L × S) where L, M, and S parameters specific for age are available at this link https://www.cdc.gov/growthcharts/percentile_data_files.htm (accessed on 28 January 2022). The distribution of weight class and WtHr ≤ 50% or >50% at baseline and at follow-up was described by bar plots and the proportions were compared by an extension of the McNemar test for the marginal homogeneity of categorical data with more than two levels [26], or by the McNemar test.

The distributions of dyslipidemia, insulin resistance and hyperuricemia at baseline and at follow-up were displayed through alluvial plots.

#### 2.7.1. Different Multiple Linear Regression Models Were Used to Assess

The impact of gender, age, puberty transition, the difference between BMI z-score at baseline and at follow-up, family history of dyslipidemia/diabetes on LDL cholesterol or HDL cholesterol or triglycerides or HOMA index or uric acid plasma values at follow-up. The models were adjusted for values of LDL cholesterol or HDL cholesterol or triglycerides or HOMA index or uric acid plasma at baseline.

#### 2.7.2. Multiple Logistic Regression Models Were Used to Assess

The impact of gender, age, puberty transition, the difference between BMI z-score at baseline and at follow-up, family history of dyslipidemia/diabetes on dyslipidemia or insulin resistance or hyperuricemia at follow-up. The models were adjusted for the presence of dyslipidemia or insulin resistance or hyperuricemia at baseline. The same linear and logistic regression models were performed including the difference between WtHr at baseline and at follow-up as covariate instead of the difference between BMI z-scores.

Statistical analyses were performed with R 4.1.2 (http://www.R-project.org) (accessed on 28 January 2022). All *p*-values were two-sided, with *p*-values < 0.05 considered statistically significant.

## 3. Results

The study involved 276 children with a mean age of 10.6 (SD = 2.3) years (56% were males). Table 1 shows the clinical characteristics and hematochemical variables of the population at baseline and at the end of follow-up.

At recruitment, 14% of children were NW, 35% were OW, and 50% were OB (Figure 1A), and 70% of subjects had a WtHr > 50% (Figure 1B). Forty-three patients (15.8%) had a family history of diabetes mellitus, and 117 (42.9%) had a family history of dyslipidemia. At least one alteration in plasma lipid values was present in 69 children (25.0%), 157 (56.9%) had HOMA index values > 90th percentile, and 46 (16.7%) had uric acid levels > 90th percentile. At the end of the follow-up period, BMI z-score and WtHr had significantly decreased (from 1.8 to 1.5 and from 53.6% to 50.7%, respectively, *p* < 0.001). The proportion of NW subjects had significantly increased, whereas the proportion of OB subjects was significantly lower than at baseline (*p* < 0.001) (Figure 1A). Only 50% of the study population had WtHr > 50% at the end of follow-up (*p* < 0.001) (Figure 1B). Changes in the prevalence of normal weight, overweight, obesity and WtHr > 50% are shown in Figure 1C,D. In detail, 40 of 237 (16.9%) children with excess weight at baseline were NW at follow-up and 63 (26.6%) had improved their weight class. In contrast, the number of subjects with a worsening of weight class was 8 out of 137 (5.8%). Regarding the prevalence of WtHr > 50%, 55 of 192 (28.6%) subjects had normalized their WtHr values (i.e., WtHr ≤ 50%) at follow-up, whereas 3 of 83 (3.6%) children with WtHr ≤ 50% at baseline showed elevated WtHr values (i.e., >50%) at follow-up. The percentage of children with HOMA index > 90th percentile at the end of follow-up was significantly lower than that at baseline (56.9% vs. 45.3%, *p* = 0.002), whereas we did not observe any significant change in the number of subjects with dyslipidemia and in those with uric acid values > 90th percentile. However, a significant reduction in LDL cholesterol levels was observed (from 95.0 to 90.4 mg/dL, *p* < 0.001).

Changes in the prevalence of dyslipidemia and elevated HOMA index and uric acid values are described in Figure 2. In detail, 32 of 69 (46.4%) children with dyslipidemia at baseline had a normal lipid profile at follow-up. In contrast, 21 of 207 (10.1%) subjects who were not dyslipidemic at baseline had dyslipidemia at follow-up. Considering the presence of insulin resistance (HOMA index > 90th percentile), 67 of 157 (42.7%) subjects with elevated HOMA index values at baseline had normal values at follow-up, whereas 35 of 119 (29.4%) children with HOMA index ≤ 90th percentile at baseline showed elevated values at follow-up. Finally, 26 of 46 (56.5%) children with elevated uric acid values (>90th percentile) at baseline had values ≤90th percentile at follow-up. In contrast, 25 of 230 (10.9%) subjects with normal uric acid values presented with hyperuricemia at follow-up (Figure 2).

Multivariate analyses showed that each one-point reduction in BMI z-score was associated with a reduction of 8.9 mg/dL in LDL cholesterol and 20.3 mg/dL in triglycerides (*p* < 0.001, Table 2), 1.63 points in the HOMA index (*p* < 0.001, Table 3) and 0.42 mg/dL in uricemia (*p* = 0.001, Table 4). The loss of one-point BMI z-score was also associated with a 77% decrease in the risk of having a HOMA index value > 90th percentile at the end of follow-up (*p* < 0.001, Table 3 and a 68% decrease in the risk of having uric acid levels >90th percentile (*p* < 0.05, Table 4). Higher baseline values of LDL cholesterol, triglycerides, HOMA index and uric acid were strongly associated with the likelihood of no improvement at follow-up (*p* < 0.001, Table 2, Table 3 and Table 4). The presence of dyslipidemia, HOMA index values > 90th percentile and uric acid values > 90th percentile was associated with a higher risk of having the same condition at the end of follow-up (*p* < 0.001, Table 2, Table 3 and Table 4). Male sex and transition from pre-pubescence to pubescence during follow-up were factors associated with higher uric acid values at follow-up (*p* = 0.01, Table 4). When the model was adjusted for modifications of eGFR between baseline and follow-up, the result did not change (data not shown).

When WtHr was included in the models instead of BMI z-score, the results were overlapping (Table 2, Table 3, and Table 4).

## 4. Discussion

Our study shows that, without the use of pharmacological therapy, an intervention based only on the modification of dietary habits and lifestyles is able not only to improve the weight status in a population of children and adolescents, but also to correct, in a large number of cases, the metabolic alterations present (alterations in the lipid profile, insulin resistance and hyperuricemia). However, in spite of the intervention, a certain number of subjects still experience an increase in BMI and a worsening of the metabolic picture, even if in a smaller percentage than those who improve.

Overall, we obtained a clear improvement in the weight status of the study population. The percentage of children with obesity was significantly reduced and, in parallel, the number of subjects with normal body weight and those who were in excessive weight but without evidence of obesity increased. The magnitude of BMI reduction was strongly associated with an improvement in the metabolic risk profile. Importantly, only a small minority of children (less than 6%) worsened their weight class, whereas a higher proportion (about 27%) experienced improvement. A Cochrane meta-analysis including 70 randomized clinical trials, based on both diet and physical activity interventions, showed that the intervention led to a significant reduction of the BMI z-score [11].

Numerically few data are available on the effects of nonpharmacological treatment on the metabolic profile of children/adolescents and, as highlighted in the meta-analysis by Ho et al., studies often analyzed only small samples of children [13]. Dyslipidemia has a non-negligible prevalence in pediatric age, especially in children and adolescents with obesity [8,27]. Isolated hypercholesterolemia is still the most frequently observed form, but the current prevalence of excess weight in younger generations is contributing to an increase in mixed forms and of those characterized only by the presence of hypertriglyceridemia [25]. In adults, the effectiveness of dietary-behavioral treatment in improving the hematochemical parameters that characterize dyslipidemia has been widely demonstrated; decreasing saturated fat and increasing fiber in the diet is associated with reductions in total and LDL cholesterol [28,29], whereas increasing physical activity and reducing free sugar consumption primarily lead to a decrease in triglycerides values [30,31]. The meta-analysis by Ho et al. [13], which included five studies, showed that lifestyle intervention had a significantly greater impact on triglycerides (−0.20 mmol/L in the short-term and −0.09 mmol/L in the long-term studies) and LDL cholesterol values (−0.30 mmol/L) than no treatment. A more recent study designed to evaluate the effect of a community lifestyle intervention (achieving healthy eating patterns and increasing physical activity) on components of the metabolic syndrome in adolescents found an improvement in triglycerides in boys and HDL cholesterol in both genders [32]. These findings are in line with those of our study, in which intervention is associated with a decrease in plasma LDL cholesterol and triglyceride values, which is closely related to the decrease in BMI z-score. Baseline LDL cholesterol and triglyceride values have an important influence on the values found at follow-up; the higher the baseline value, the higher the final value. Although the prevalence of a family history of dyslipidemia is high in our sample (43%), having one or two dyslipidemic parents is not a determining factor for the response to treatment. It is possible to hypothesize that the altered lipid values in this population are largely due to poor dietary habits in the family, which causes an altered lipid picture in both children and their parents, rather than to the actual presence of a familial dyslipidemia.

The meta-analysis of Ho et al. [13] included four studies that reported results on insulin resistance: the difference in HOMA index was −2.32 (95% CI: −3.25 to −1.39) in favor of lifestyle intervention compared to the control group, However, the heterogeneity (among these studies) was high. In addition, a meta-analysis that analyzed the ability of exercise training to lower insulin resistance in children and/or adolescents classified with obesity or as being overweight showed a significant reduction of HOMA index (−0.61, 95% CI: −1.19 to −0.02) [14]. However, a study designed to evaluate the effects of recommendations to follow the DASH diet vs. a usual diet on the characteristics of metabolic syndrome in Iranian adolescents resulted in a decrease in serum insulin levels (101.4 vs. 90 pmol/L, *p* = 0.04) without any significant reduction in HOMA index in the DASH group [33]. A gradual decline in insulin secretion from normal glucose tolerance to impaired glucose tolerance to type 2 diabetes mellitus in adolescents with obesity has been documented [34]. A high HOMA index was the most frequent metabolic alteration in our sample; in fact, more than half of children had a HOMA index > 90th percentile. It is known that in pediatric ages, insulin resistance increases physiologically with age and, in particular, with pubertal development [35]. In our population, the proportion of prepubertal subjects decreased from 58 to 28% from baseline to follow-up, and this was accompanied, as expected, by an increase in mean HOMA index values. However, we observed a significant reduction in HOMA index values in children/adolescents who decreased in body weight: for each lost BMI z-score point, the HOMA index was reduced by 1.6. In addition, the likelihood of insulin resistance (i.e., HOMA index > 90th percentile) decreased by 77%. Although the presence of high HOMA index values at baseline strongly conditions the outcome, it seems clear that this parameter responds favorably to nonpharmacological treatment. We might have expected that the presence of a diabetic parent would have an impact on the presence and/or persistence of insulin resistance in the offspring. However, familiarity of diabetes mellitus did not influence the results regarding HOMA index, neither when analyzed as a continuous variable nor as a categorical variable. This finding is similar to what we observed for a family history of dyslipidemia; having a parent with dyslipidemia had no significant impact either on the presence of dyslipidemia at baseline or at follow-up in children. In a small sample of girls with obesity, Browning et al. evaluated the difference in some metabolic variables after six months of an intervention combining diet, behavioral counseling, and exercise training. In patients who lost weight (slightly more than half of the study population), the authors observed a reduction in LDL cholesterol (but not triglycerides) and basal blood glucose compared with those who gained weight [36].

In our population, the number of children with a WtHr > 50% significantly decreased after the intervention. Approximately 29% of subjects with a WtHr > 50% at baseline had a WtHr < 50% at the end of the follow-up period. Replacing BMI z-score by WtHr (visceral fat index) in the statistical models had no additional effect on outcomes, although from a theoretical point of view, waist circumference should be better associated with metabolic alterations than BMI z-score. However, our result confirms the validity of WtHr in determining pediatric cardiovascular risk, but also suggests that this parameter does not provide any substantial advantage over BMI. It is possible to assume, however, that more accurate estimates of central adiposity (such as the use of dual-energy x-ray absorptiometry) could have led to different results. In a small sample of Japanese children, dietary intervention combined with exercise treatment led to a reduction in the areas of subcutaneous and visceral fat, quantified in CT images obtained by a body fat scan, in association with a significant decrease in plasma values of total cholesterol, triglycerides and insulin [37]. Abdominal-visceral obesity is associated with unfavorable metabolic activity and increased cardiovascular risk. The metabolic activity of visceral fat is a key factor in the development of obesity-related complications [38]. Visceral obesity results in a chronic inflammatory state in adipose tissue. Visceral adipose tissue accumulation leads to immune cell infiltration, release of vasoconstrictor mediators, dysfunctional remodeling and fibrosis of with increased vascular stiffness [39]. Therefore, fat distribution more than BMI may determine cardiovascular disease risk. Although some individuals may apparently accumulate excess body fat without developing cardiometabolic diseases [40], it has been shown that a non-negligible proportion of “metabolically healthy” children with obesity have elevated values of non-traditional risk factors such as HOMA index and serum uric acid [41].

In children, as well as in adults, an association has been demonstrated between obesity and uric acid levels [10,42,43]. There is no agreement on which of the two factors determines the other, and some evidence suggests that this relationship may be bidirectional [44]. From our data, it appears that hyperuricemia is the variable that is least responsive to nonpharmacologic treatment, as the percentage of children with uric acid > 90th percentile at baseline and at follow-up is overlapping. It should be noted, however, that half of the subjects with hyperuricemia at baseline are different from those with hyperuricemia at follow-up. A recent longitudinal study evaluating the temporal relationship between BMI and uric acid showed that changes in BMI preceded changes in uric acid, providing evidence of a causal relationship between obesity and hyperuricemia [45]. Our data, showing a significant association between reduction of BMI z-score (and WtHr) and uric acid values at follow-up, seem to support this hypothesis and also suggest that, at least in pediatric ages, if an increase in weight leads to an increase in uric acid, its reduction can reverse at least in part the phenomenon. Another recent study performed in a population of children and adolescents with obesity demonstrated a reduction in uric acid levels in children who underwent a multifactorial lifestyle intervention and lost weight during the trial, and an increase in those who gained weight [46]. In our population, the higher the uric acid values at baseline, the lower the response to the intervention, and children with uric acid values > 90th percentile at baseline were six times more likely to be hyperuricemic even at follow-up. It has been shown that children with higher levels of uricemia are those in whom it is more difficult to obtain a benefit from the intervention also in terms of reduction of blood pressure values, a parameter associated with high levels of uric acid also in pediatric age [47]. An association between uric acid values and intake of sugar-sweetened beverages (SSBs) has been robustly demonstrated in children [48,49]. However, studies that have investigated the impact of measures to reduce SSBs intake have not explored the effect of the intervention on plasma uric acid, but only on anthropometric parameters [50,51,52,53]. The factors that influence uric acid values are complex and not only depend on weight and diet, but also on the genetic characteristics of the subject. It is therefore possible that uric acid values are not reduced in all children in the same way, even if a similar decrease in BMI z-score is obtained. As expected, eGFR decreased with increasing age of the study sample. However, no child/adolescent showed creatinine/eGFR values outside the normal range. It has been shown that creatinine values increase progressively from the first years of life, in accordance with the increase of muscle mass with consequent reduction of eGFR values [54]. Since this is a physiological phenomenon that occurs in all subjects and since BMI has been indexed for age (z-score) in all statistical models, we think that, in the absence of renal disease, the physiological reduction of eGFR should not influence our results.

Vegetarian diets in adults are assumed to be healthy; however, it is unclear whether this is also true for children and adolescents. So-called “plant-diets” have been shown to have a protective effect on the risk of type 2 diabetes, cardiovascular disease, and obesity in adults [55]. Existing data do not allow to reach a solid conclusion on the health benefits or risks of vegetarian diets on the nutritional and health status of children and adolescents [56]. A vegan diet can potentially be critical in children because of inadequate intake of protein, long-chain fatty acids, iron, zinc, vitamin D, iodine, calcium, and in particular vitamin B12 [57]. However, some scientific societies allow it, but only if properly integrated and controlled [58]. In our study we chose to propose a Mediterranean diet, as it is closer to the culture of Italy, including specific indications based on metabolic alterations present in individual cases. Moreover, we have always prescribed an increase in the consumption of fruits and vegetables.

A strength of our study is that, whereas most previously published studies on this topic were limited to analyzing the differences between pre- and post-intervention values in children who lose weight compared with those who do not, we also provide a precise estimate of the effect of our intervention. For each point of BMI z-score lost, we described a reduction of about 9 mg/dL of LDL cholesterol, 20 mg/dL of triglycerides, 1.6 of HOMA index and 0.4 mg/dL of uric acid. In the available studies, anthropometric variables were not always indexed, and the influence that age and puberty have on lipid and glucose patterns and uric acid values in developmental age was often not considered. In our study, all variables were analyzed as both continuous and indexed variables, and all models were adjusted for the onset of puberty.

A weakness of our study is that we could detect only one indirect parameter of adiposity (waist circumference). A more correct estimation of visceral fat and its changes in relation to those of hematochemical parameters could give interesting additional information. Moreover, as our society has become increasingly multiethnic, other factors, such as sociocultural influences, may contribute, beyond biological factors, to the success of non-pharmacological treatment directed at a population of children and adolescents with excess weight. The presence of community workers specifically trained to promote healthy behaviors that take into account cultural background, psychosocial stressors, and economic limitations would likely have improved compliance among our population and allowed for better outcomes. Unfortunately, our center is not equipped with this kind of support.

## 5. Conclusions

In conclusion, in children it is not easy to evaluate the effect of a nonpharmacological treatment on metabolic variables that may represent a cardiovascular risk factor. In fact, children are growing organisms in which these parameters change physiologically. Moreover, body weight, a factor strongly associated with metabolic parameters, undergoes important modifications at this age. By using models that correct these confounding factors, our study shows that a change in lifestyle and dietary habits is able to induce an improvement in weight, lipid profile, glycemic and uric acid values. For all metabolic parameters considered, baseline values are strongly associated with those found at follow-up. This means that the more a child presents a metabolic impairment, the more difficult it will be, with the same weight reduction, to bring back its parameters in a normal range. These considerations confirm and emphasize that an early intervention aimed at normalizing the lipid, glycemic and uric acid values is essential for the realization of a real prevention of cardiovascular disease in a world where this is the first cause of death.

## Figures and Tables

**Figure 1 nutrients-14-01034-f001:**
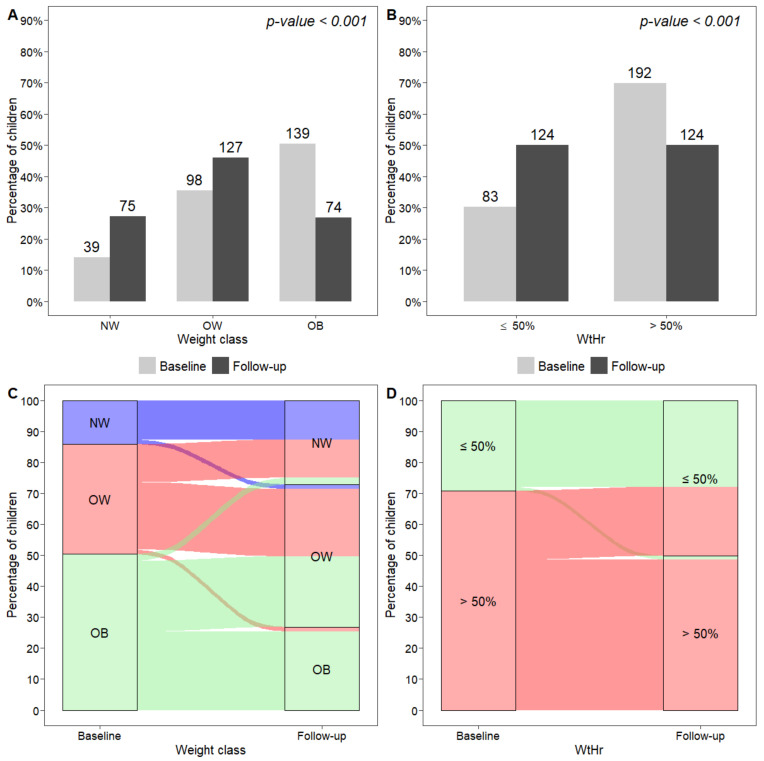
Distribution of weight classes (**A**) and WtHr ≤ 50% or >50% (**B**) at baseline and at follow-up, and graphical representation of subjects moving from one weight class to another (**C**), and from a WtHr value > 50% to a value ≤50% and vice versa (**D**), from baseline to follow-up. NW, normal weight; OW, overweight; OB, obese; WtHr, waist-to-height-ratio.

**Figure 2 nutrients-14-01034-f002:**
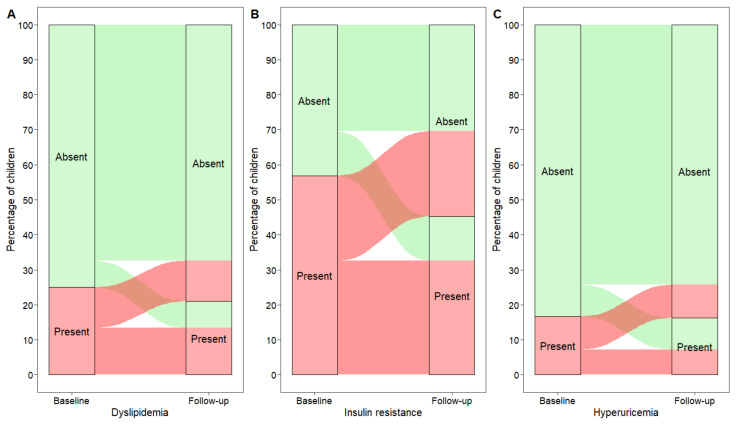
Graphical representation of subjects moving from a pathological to a normal clinical condition and vice versa ((**A**) Dyslipidemia; (**B**) Insulin resistance; (**C**) Hyperuricemia), from baseline to follow-up.

**Table 1 nutrients-14-01034-t001:** Anthropometric and clinical characteristics at baseline and at follow-up.

Parameter	Baseline (*n* = 276)	Follow-Up (*n* = 276)	*p*
Age (years), mean (SD)	10.6 (2.3)	12.1 (2.3)	<0.001
Gender (males), n (%)	154 (55.8)	-	
Puberty, n (%)	115 (42.0)	179 (71.6)	<0.001
Weight (kg), median (IQR)	51.4 (40.0, 62.5)	55.9 (44.4, 68.1)	<0.001
Height (cm), mean (SD)	145.3 (14.3)	153.1 (13.5)	<0.001
BMI (kg/m^2^), mean (SD)	24.2 (4.5)	23.8 (4.5)	0.004
BMI (z-score), median (IQR)	1.8 (1.3, 2.1)	1.5 (1.0, 1.8)	<0.001
Waist circumference (cm), mean (SD) ^§^	77.9 (12.1)	77.6 (11.7)	0.125
WtHr (%), mean (SD)	53.6 (6.9)	50.7 (6.6)	<0.001
Family history of diabetes (mother and/or father), n (%)	43 (15.8)	-	
Family history of dyslipidemia (mother and/or father), n (%)	117 (42.9)	-	
Total cholesterol (mg/dL), mean (SD)	163.9 (28.3)	160.4 (28.4)	0.002
HDL cholesterol (mg/dL), median (IQR)	52.0 (46.0, 60.0)	53.0 (45.0, 62.0)	0.335
HDL cholesterol < 40 mg/dL, n (%)	31 (11.2)	30 (10.9)	0.999
LDL cholesterol (mg/dL), median (IQR)	95.0 (78.8, 111.7)	90.4 (74.9, 106.2)	<0.001
LDL ≥ 130 mg/dL, n (%)	28 (10.1)	16 (5.8)	0.010
Triglycerides (mg/dL), median (IQR)	65.0 (49.0, 88.3)	66.0 (47.0, 88.0)	0.967
Triglycerides ≥ 100 mg/dL or ≥130 mg/dL ^#^, n (%)	27 (9.8)	24 (8.7)	0.735
Dyslipidemia, n (%)	69 (25.0)	58 (21.0)	0.170
Glucose (mg/dL), mean (SD)	83.1 (7.6)	83.9 (8.4)	0.118
Insulin (µU/mL), median (IQR)	12.1 (7.7, 16.0)	10.8 (7.32, 14.9)	0.925
HOMA Index ^$^, median (IQR)	2.5 (1.5, 3.4)	2.2 (1.5, 3.1)	0.480
HOMA Index > 90th percentile, n (%)	157 (56.9)	125 (45.3)	0.002
Uric acid (mg/dL), mean (SD)	4.4 (1.1)	4.6 (1.2)	<0.001
Uric acid > 90th percentile, n (%)	46 (16.7)	45 (16.3)	1.000
Creatinine (mg/dL), mean (SD)	0.5 (0.1)	0.6 (0.1)	<0.001
eGFR (mL/min), mean (SD)	155.5 (25.5)	146.3 (24.2)	<0.001
Follow-up time (months), median (IQR)	14.7 (12.4, 19.3)	-	

SD, standard deviation; IQR, interquartile range. Comparisons were conducted through the *t*-test for paired data or the Wilcoxon signed-rank test for continuous variables, and by the McNemar test for categorical variables. BMI, body mass index; WtHr, waist-to-height-ratio; eGFR, estimated glomerular filtration rate. ^#^ Triglycerides ≥ 100 if children < 10 years or ≥130 if children ≥ 10 years. ^$^ Calculated as plasma insulin (µU/mL) × plasma glucose (mmol/L)/22.5. ^§^ missing at baseline *n* = 1, missing at follow-up *n* = 28.

**Table 2 nutrients-14-01034-t002:** Effect of LDL, HDL or triglycerides at baseline, gender, transition from pre-pubescent to pubescent, BMI or WtHr, family history of dyslipidemia on LDL, HDL or triglycerides at follow-up, by multiple linear regression models. Effect of dyslipidemia at baseline, gender, transition from pre-pubescent to pubescent, BMI or WtHr, family history of dyslipidemia on dyslipidemia at follow-up, by a multiple logistic regression model.

LDL Cholesterol at Follow-Up							
Variable	b	(95% CI)	*p*	Variable	b	(95% CI)	*p*
Intercept	23.167	(15.135; 31.200)	<0.001	Intercept	22.264	(14.359; 30.169)	<0.001
LDL cholesterol at baseline	0.743	(0.668; 0.818)	<0.001	LDL cholesterol at baseline	0.754	(0.679; 0.829)	<0.001
Gender (males)	−3.151	(−7.017; 0.715)	0.110	Gender (males)	−3.695	(−7.618; 0.228)	0.065
Becoming pubescent during follow-up	1.809	(−2.550; 6.168)	0.414	Becoming pubescent during follow-up	2.815	(−1.618; 7.249)	0.212
BMI (Δz-scores)	−8.869	(−14.152; −3.586)	0.001	ΔWtHr	−0.803	(−1.226; −0.379)	<0.001
Family history of dyslipidemia	3.234	(−0.751; 7.219)	0.111	Family history of dyslipidemia	2.335	(−1.677; 6.346)	0.253
**HDL Cholesterol at Follow-Up**							
**Variable**	**b**	**(95% CI)**	** *p* **	**Variable**	**b**	**(95% CI)**	** *p* **
Intercept	16.049	(10.873; 21.224)	<0.001	Intercept	16.643	(11.573; 21.713)	<0.001
HDL cholesterol at baseline	0.718	(0.628; 0.808)	<0.001	HDL cholesterol at baseline	0.718	(0.628; 0.808)	<0.001
Gender (males)	−1.723	(−3.955; 0.509)	0.130	Gender (males)	−1.581	(−3.871; 0.709)	0.175
Becoming pubescent during follow-up	−0.379	(−2.885; 2.127)	0.766	Becoming pubescent during follow-up	−0.997	(−3.572; 1.578)	0.446
BMI (Δz-scores)	1.965	(−1.090; 5.020)	0.206	ΔWtHr	0.098	(−0.147; 0.343)	0.431
Family history of dyslipidemia	−0.318	(−2.547; 1.912)	0.779	Family history of dyslipidemia	−0.299	(−2.570; 1.973)	0.796
**Triglycerides at Follow-Up**							
**Variable**	**b**	**(95% CI)**	** *p* **	**Variable**	**b**	**(95% CI)**	** *p* **
Intercept	37.394	(27.145; 47.642)	<0.001	Intercept	34.731	(24.725; 44.737)	<0.001
Triglycerides at baseline	0.570	(0.466; 0.673)	<0.001	Triglycerides at baseline	0.544	(0.442; 0.645)	<0.001
Gender (males)	−0.865	(−7.958; 6.228)	0.810	Gender (males)	−0.390	(−7.457; 6.676)	0.913
Becoming pubescent during follow-up	3.827	(−4.161; 11.815)	0.346	Becoming pubescent during follow-up	6.997	(−0.967; 14.961)	0.085
BMI (Δz-scores)	−20.366	(−30.046; −10.687)	<0.001	ΔWtHr	−1.212	(−1.968; −0.456)	0.002
Family history of dyslipidemia	0.379	(−6.754; 7.513)	0.917	Family history of dyslipidemia	0.379	(−6.680; 7.438)	0.916
**Dyslipidemia at Follow-Up**							
**Variable**	**OR**	**(95% CI)**	** *p* **	**Variable**	**OR**	**(95% CI)**	** *p* **
Dyslipidemia at baseline	11.187	(5.523; 23.674)	<0.001	Dyslipidemia at baseline	11.418	(5.570; 24.509)	<0.001
Gender (males)	1.343	(0.654; 2.817)	0.426	Gender (males)	1.210	(0.580; 2.571)	0.614
Becoming pubescent during follow-up	1.085	(0.197; 1.497)	0.841	Becoming pubescent during follow-up	1.200	(0.521; 2.686)	0.661
BMI (Δz-scores)	0.557	(0.436; 1.885)	0.255	ΔWtHr	0.939	(0.864; 1.014)	0.120
Family history of dyslipidemia	0.915	(0.436; 1.885)	0.812	Family history of dyslipidemia	0.877	(0.410; 1.837)	0.730

b indicates multivariate coefficient; CI, confidence interval; BMI, body mass index; WtHr, waist-to-height-ratio; OR, odds ratio; Δ indicates the difference between the baseline value and the follow-up value.

**Table 3 nutrients-14-01034-t003:** Effect of HOMA index at baseline, gender, transition from pre-pubescent to pubescent, BMI or WtHr, family history of diabetes on HOMA index at follow-up by a multiple linear regression model. Effect of insulin resistance at baseline, gender, transition from pre-pubescent to pubescent, BMI or WtHr, family history of diabetes on insulin resistance at follow-up by a multiple logistic regression model.

HOMA Index at Follow-Up							
Variable	b	(95% CI)	*p*	Variable	b	(95% CI)	*p*
Intercept	1.775	(1.255; 2.295)	<0.001	Intercept	1.311	(0.785; 1.836)	<0.001
HOMA index at baseline	0.495	(0.385; 0.605)	<0.001	HOMA index at baseline	0.502	(0.386; 0.618)	<0.001
Gender (males)	−0.128	(−0.557; 0.301)	0.558	Gender (males)	−0.030	(−0.479; 0.418)	0.894
Becoming pubescent during follow-up	0.377	(−0.112; 0.866)	0.130	Becoming pubescent during follow-up	0.562	(0.051; 1.073)	0.031
BMI (Δz-scores)	−1.637	(−2.228; −1.047)	<0.001	ΔWtHr	−0.072	(−0.121; −0.024)	0.004
Family history of diabetes	0.098	(−0.495; 0.690)	0.746	Family history of diabetes	0.151	(−0.474; 0.777)	0.634
**Insulin Resistance (HOMA Index > 90th Percentile) at Follow-Up**							
**Variable**	**OR**	**(95% CI)**	** *p* **	**Variable**	**OR**	**(95% CI)**	** *p* **
Insulin resistance at baseline	4.055	(2.299; 7.340)	<0.001	Insulin resistance at baseline	3.716	(2.116; 6.683)	<0.001
Gender (males)	1.752	(1.009; 3.073)	0.048	Gender (males)	1.654	(0.952; 2.900)	0.076
Becoming pubescent during follow-up	1.140	(0.608; 2.138)	0.682	Becoming pubescent during follow-up	1.296	(0.689; 2.449)	0.421
BMI (Δz-scores)	0.227	(0.097; 0.503)	<0.001	ΔWtHr	0.951	(0.894; 1.009)	0.102
Family history of diabetes	0.613	(0.282; 1.295)	0.205	Family history of diabetes	0.514	(0.227; 1.116)	0.100

b indicates multivariate coefficient; CI, confidence interval; BMI, body mass index; WtHr, waist-to-height-ratio; OR, odds ratio; Δ indicates the difference between the baseline value and the follow-up value.

**Table 4 nutrients-14-01034-t004:** Effect of uric acid at baseline, gender, transition from pre-pubescent to pubescent, BMI or WtHr on uric acid at follow-up by a multiple linear regression model. Effect of hyperuricemia at baseline, gender, transition from pre-pubescent to pubescent, BMI or WtHr on hyperuricemia at follow-up by a multiple logistic regression model.

Uric Acid at Follow-Up							
Variable	b	(95% CI)	*p*	Variable	b	(95% CI)	*p*
Intercept	1.087	(0.710; 1.464)	<0.001	Intercept	1.016	(0.627; 1.406)	<0.001
Uric acid at baseline	0.766	(0.688; 0.845)	<0.001	Uric acid at baseline	0.770	(0.688; 0.852)	<0.001
Gender (males)	0.307	(0.130; 0.485)	0.001	Gender (males)	0.293	(0.108; 0.477)	0.002
Becoming pubescent during follow-up	0.259	(0.058; 0.460)	0.012	Becoming pubescent during follow-up	0.293	(0.084; 0.503)	0.006
BMI (Δz-scores)	−0.421	(−0.664; −0.177)	0.001	ΔWtHr	−0.028	(−0.048; −0.008)	0.006
**Hyperuricemia (Uric Acid > 90th Percentile) at Follow-Up**							
**Variable**	**OR**	**(95% CI)**	** *p* **	**Variable**	**OR**	**(95% CI)**	** *p* **
Hyperuricemia at baseline	6.236	(2.803; 12.101)	<0.001	Hyperuricemia at baseline	6.052	(2.641; 14.117)	<0.001
Gender (males)	0.821	(0.395; 1.715)	0.597	Gender (males)	0.710	(0.332; 1.514)	0.374
Becoming pubescent during follow-up	2.157	(0.970; 4.758)	0.056	Becoming pubescent during follow-up	2.651	(1.165; 6.040)	0.019
BMI (Δz-scores)	0.318	(0.097; 0.952)	0.048	ΔWtHr	0.920	(0.842; 1.000)	0.059

b indicates multivariate coefficient; CI, confidence interval; BMI, body mass index; WtHr, waist-to-height-ratio; OR; odds ratio; Δ indicates the difference between the baseline value and the follow-up value.

## Data Availability

Data are available on reasonable request from the corresponding author.

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
