# Peer review of "Impact of Lifestyle Modifications on Alterations in Lipid and Glycemic Profiles and Uric Acid Values in a Pediatric Population"

_nutrients, 2022, doi:10.3390/nu14051034_

Round 1
Reviewer 1 Report
This manuscript reports on the effects of non-pharmacological treatment interventions on lipid and glycemic profiles in children. The data provided add to the knowledge base for this research field and could be considered for publication after the authors improve upon the submitted manuscript. Further, the manuscript would benefit from a language review by a native English speaker if the version provided was fully checked for language by the authors. A few examples have been addressed in the details below, but this list is not complete. Please see details below for manuscript revisions:
ABSTRACT:
- L14: Here is an example for the language revisions: “The aim of the study was to evaluate the effects…” you are assessing more than one effect.
- L15: You should include the range of age of your population. Pediatric is still a quite general term.
- L17: Please state the sample matrix that was used for biochemical analysis, as shown later it is plasma/serum.
- L17,18: sentence should be revised for grammar.
- L19: The median follow up should be accompanied by the IQR or range.
- L20: I understand that the abstract is very limited in word count, but “non-pharmacological treatment” should be a bit more specified as it is a focus of your manuscript. L22ff: all values presented should be explained. I assume we are looking at medians? Or means? Also, you should add the IQR or SD values as well.
- L24: omit “also”.
- L26: If you state that “Improvement of dietary habits and lifestyles can correct metabolic alterations” the reader may think that these interventions will complete erase the alterations in lipid and glycemic profiles. Based on your data I would recommend to state upon these interventions, the profiles have improved.
INTRODUCTION:
- L41: “… blood pressure values in children…”
- Your introduction should include a quick section in which you establish the biomarkers of interest as typical biomarkers for monitoring of changes in metabolic profiles in your co-hort of children.
MATERIALS AND METHODS:
- L52-60: please ad the number of participants.
- L54: Please specify the location of your Unit for Cardiovascular Risk Assessment
- L61ff: The next 3 sections (Baseline and follow-up assessments to Biochemical parameters) is a bit muddled. Clearly define what you need to put under each sub-header and do not repeat already stated information. If this is how you want to separate the paragraphs, that’s fine, but then the first one should only explain how baseline and follow-up was conducted, without including information on the other two sub-headers, the second one explains anthropometric, and the last one explains the biochemical markers.
- L61-69: This paragraph should include more details regarding the timing. How much time between base line and follow up? The follow up describes their last visit with the Unit? Measurements were only taken at baseline and at one follow up, not during the 3 months visits? How many of the periodic visits were scheduled?
- L70-81: Please explain how the primary anthropometric measurements in the same sequence than stated in the beginning of this section. Then move on to the secondary outcomes, such as BMI (secondary as it is derived from your primary measures).
- L74: the link to the CDC website does not bring up the charts as indicated. Don’t provide a general link to the vicinity of what you were using. Please explain what was done and if links are added, they need to get the reader exactly where these charts are. The reader needs to be enabled by you to follow your approach. If there is an equation involved, it should be included in your manuscript, too. Important details are missing here.
- L74ff: Weight class classifications should be added, not just the reference. How is NW, OW, OB defined? These are again primary outcomes; this needs to be included in more details.
- L82-92f: The stated measurements of biochemical markers are not matching what you have stated in the abstract or introduction. State the complete list of analyses at the beginning of the paragraph, and explain to why certain biomarkers are mentioned here, but not before. Explain in more detail how each biomarker was measured. What assay kit was used by which manufacturer? If it’s not a commercial assay, add details about the analyses including a reference if available. The current description is not sufficient.
RESULTS:
- Table 1: please state the statistical analyses used for the data presented in the table footnote.
- Figure 1: labels on the y-axis for panels A and B are missing. What does the p-value refer to in panels A and B? Overall or pairwise comparison? Pairwise within weight group would be good to add. Panels C and D look nice but I don’t see the value they bring to the paper? The reader does not get the actual values but only see that smaller percentages switched weight class or WtHr. Thus, the data should be presented in actual values, which then renders these panels as redundant.
- Figure 2: Please see comment about Figure 1 panels C and D.
- Generally, don’t present data repetitively, chose one format or the other (table or figure).
- Table 2: Why are using “Panel A” and “Panel B” as column header? Just use the actual term as indicated in the description, which will be a lot easier on the reader. Since the point of the manuscript is for others to read and understand you should be aiming for an easy-to-understand manuscript to readers who are less familiar with the work you do.
DISCUSSION:
- L305: Please provide an example of sample size to define “small samples”
- L386: “support” instead of “validate”
Author Response
Reviewer 1
We thank the reviewer for his/her suggestions that allowed us to improve our manuscript.
Comments and Suggestions for Authors
This manuscript reports on the effects of non-pharmacological treatment interventions on lipid and glycemic profiles in children. The data provided add to the knowledge base for this research field and could be considered for publication after the authors improve upon the submitted manuscript. Further, the manuscript would benefit from a language review by a native English speaker if the version provided was fully checked for language by the authors. A few examples have been addressed in the details below, but this list is not complete. Please see details below for manuscript revisions:
ABSTRACT:
- L14: Here is an example for the language revisions: “The aim of the study was to evaluate the effects…” you are assessing more than one effect.
The sentence has been corrected as suggested by the reviewer.
- L15: You should include the range of age of your population. Pediatric is still a quite general term.
The age range (SD) of the population is already in the abstract on the next line: “The study involved 276 subjects with a mean age of 10.6 (2.3) years”.
- L17: Please state the sample matrix that was used for biochemical analysis, as shown later it is plasma/serum.
The sample matrix used for biochemical analysis was serum.
- L17,18: sentence should be revised for grammar.
The sentence has been rewritten as requested by the reviewer.
- L19: The median follow up should be accompanied by the IQR or range.
The range of follow-up months has been added.
- L20: I understand that the abstract is very limited in word count, but “non-pharmacological treatment” should be a bit more specified as it is a focus of your manuscript.
As requested by the reviewer this point has been better specified.
- L22ff: all values presented should be explained. I assume we are looking at medians? Or means? Also, you should add the IQR or SD values as well.
Values presented are those from multivariate models. Confidence intervals have been added as requested by the reviewer.
- L24: omit “also”.
The word "also" was deleted.
- L26: If you state that “Improvement of dietary habits and lifestyles can correct metabolic alterations” the reader may think that these interventions will complete erase the alterations in lipid and glycemic profiles. Based on your data I would recommend to state upon these interventions, the profiles have improved.
We agree with the reviewer's observation. The sentence has been corrected as suggested.
INTRODUCTION:
- L41: “… blood pressure values in children…”
The sentence has been corrected as suggested by the reviewer.
- Your introduction should include a quick section in which you establish the biomarkers of interest as typical biomarkers for monitoring of changes in metabolic profiles in your co-hort of children.
As requested by the reviewer a sentence to highlight which biomarkers were chosen was added to the text.
MATERIALS AND METHODS:
- L52-60: please ad the number of participants.
The number of participants has been added
- L54: Please specify the location of your Unit for Cardiovascular Risk Assessment
The location of our Cardiovascular Risk Assessment Unit has been specified.
- L61ff: The next 3 sections (Baseline and follow-up assessments to Biochemical parameters) is a bit muddled. Clearly define what you need to put under each sub-header and do not repeat already stated information. If this is how you want to separate the paragraphs, that’s fine, but then the first one should only explain how baseline and follow-up was conducted, without including information on the other two sub-headers, the second one explains anthropometric, and the last one explains the biochemical markers.
The three sections have been modified as requested by the reviewer.
- L61-69: This paragraph should include more details regarding the timing. How much time between base line and follow up? The follow up describes their last visit with the Unit? Measurements were only taken at baseline and at one follow up, not during the 3 months visits? How many of the periodic visits were scheduled?
Information requested by the reviewer was added to the manuscript (Section “Baseline and follow-up assessments”).
- L70-81: Please explain how the primary anthropometric measurements in the same sequence than stated in the beginning of this section. Then move on to the secondary outcomes, such as BMI (secondary as it is derived from your primary measures).
The paragraph has been rearranged as requested by the reviewer.
- L74: the link to the CDC website does not bring up the charts as indicated. Don’t provide a general link to the vicinity of what you were using. Please explain what was done and if links are added, they need to get the reader exactly where these charts are. The reader needs to be enabled by you to follow your approach. If there is an equation involved, it should be included in your manuscript, too. Important details are missing here.
As requested by the reviewer the link that allows direct access to the CDC charts has been added: https://www.cdc.gov/growthcharts/clinical_charts.htm. Through this link the reader can easily access the data table of BMI for age Charts. The Z-score value was calculated by the equation (BMI value/M)^L-1)/(L*S) where L, M, S parameters specific for age are available at this link https://www.cdc.gov/growthcharts/percentile_data_files.htm. This information was added to the manuscript (Section “Statistical analysis”).
- L74ff: Weight class classifications should be added, not just the reference. How is NW, OW, OB defined? These are again primary outcomes; this needs to be included in more details.
The definition of the IOTF has been widely accepted and used for many years both in the clinical field and in pediatric research. It is based on an international survey by Cole (ref 15) who proposed, based on data derived from six large nationally representative growth studies, specific cut-off points for age and gender, from 2 to 18 years, to define overweight and obesity. For each of the surveys, centile curves were drawn that at age 18 years passed through the widely used cut off points of 25 and 30 kg/m2 for adult overweight and obesity. We believe that this classification is known to most readers and that it is sufficient to cite the manuscript. However, if the reviewer considers it necessary, we can explain in detail what methodology Cole used to construct his classification.
- L82-92f: The stated measurements of biochemical markers are not matching what you have stated in the abstract or introduction. State the complete list of analyses at the beginning of the paragraph, and explain to why certain biomarkers are mentioned here, but not before. Explain in more detail how each biomarker was measured. What assay kit was used by which manufacturer? If it’s not a commercial assay, add details about the analyses including a reference if available. The current description is not sufficient.
Commercial kits, normally used for routine examinations of our patients, were employed for all analyses. All details requested by the reviewer have been added to the section “Biochemical parameters”.
RESULTS:
- Table 1: please state the statistical analyses used for the data presented in the table footnote.
The requested information has been added in the table footnote.
- Figure 1: labels on the y-axis for panels A and B are missing.
Labels on the y-axis have been added to the Figure 1.
What does the p-value refer to in panels A and B? Overall or pairwise comparison?
P-value refer to overall comparison.
Pairwise within weight group would be good to add.
We showed in Figure 1, Panels A and B only the p-value of the overall comparison obtained through an extension if the McNemar test for paired data since it gives information about the marginal homogeneity of categorical data in the presence of more than 2 categories.
Panels C and D look nice but I don’t see the value they bring to the paper? The reader does not get the actual values but only see that smaller percentages switched weight class or WtHr. Thus, the data should be presented in actual values, which then renders these panels as redundant.
We prefer not to remove panels C and D since we think that they give a clear representation of how our population change the weight class and the WtHr level during the follow-up.
Figure 2: Please see comment about Figure 1 panels C and D.
We prefer not to remove this figure since we think that it gives a clear representation of how our population change according to the three outcomes: dyslipidemia, insulin resistance and hyperuricemia. Changes in the prevalence of normal weight, overweight and obesity from baseline to follow-up are described in the Results Section.
- Generally, don’t present data repetitively, chose one format or the other (table or figure).
- Table 2: Why are using “Panel A” and “Panel B” as column header? Just use the actual term as indicated in the description, which will be a lot easier on the reader. Since the point of the manuscript is for others to read and understand you should be aiming for an easy-to-understand manuscript to readers who are less familiar with the work you do.
The definition Panel A and Panel B have been removed from Tables 2,3 and 4 and from the Results Section as requested by the reviewer.
DISCUSSION:
- L305: Please provide an example of sample size to define “small samples”
The studies included in the meta-analysis analyzed samples consisting of 27 to 54 children. Only two studies had a somewhat larger sample size (Huang SH et al, Acta Paediatr Taiwan. 2007;48(4):201-206 and Savoye M et al, JAMA. 2007;297:2697-2704; n=120 and n=135, respectively).
- L386: “support” instead of “validate”
The sentence has been edited as requested by the reviewer.

Reviewer 2 Report
The authors aimed to evaluate the effects of dietary-behavioral treatment on changes in lipid and glucose profiles and uric acid levels in children and adolescents referred to a second-level outpatient clinic. The authors studied a cohort of children and adolescents (4-18 years) referred to their unit by primary care physicians because of excess weight and/or abnormal lipid and/or glucose profiles and/or to a second-level abnormal uric acid values with appropriate exclusion criteria.
Lifestyle modification has long been the first line treatment for overweight and obesity as it yields favorable effects on weight reduction and metabolic profiles, which are usually used as parameters to evaluate and follow patients with overweight and obesity in clinical settings.
Uric acid is, in particular, not routinely measured - especially in children, therefore, interesting and worth recording. The author, though, did not discuss whether the significant differences of creatinine and GFR between baseline and follow-up might/might not have effects on uric acid levels.
Minor punctuation needs attention.
Table 2 panel B for HDL and triglyceride: variable - LDL cholesterol at baseline - should these not be HDL and triglyceride respectively?
Discussion:
- Significant differences of creatinine and GFR between baseline and follow-up presented should be discussed.
- Line 353 to 355 - "Similarly, to what observed..." needs clarification.
Author Response
Reviewer 2
We thank the reviewer for his/her suggestions that allowed us to improve our manuscript.
Comments and Suggestions for Authors
The authors aimed to evaluate the effects of dietary-behavioral treatment on changes in lipid and glucose profiles and uric acid levels in children and adolescents referred to a second-level outpatient clinic. The authors studied a cohort of children and adolescents (4-18 years) referred to their unit by primary care physicians because of excess weight and/or abnormal lipid and/or glucose profiles and/or to a second-level abnormal uric acid values with appropriate exclusion criteria.
Lifestyle modification has long been the first line treatment for overweight and obesity as it yields favorable effects on weight reduction and metabolic profiles, which are usually used as parameters to evaluate and follow patients with overweight and obesity in clinical settings.
Uric acid is, in particular, not routinely measured - especially in children, therefore, interesting and worth recording. The author, though, did not discuss whether the significant differences of creatinine and GFR between baseline and follow-up might/might not have effects on uric acid levels.
Minor punctuation needs attention.
Table 2 panel B for HDL and triglyceride: variable - LDL cholesterol at baseline - should these not be HDL and triglyceride respectively?
The reviewer is absolutely right. We apologize for the error and thank him/her for the comment.
Discussion:
- Significant differences of creatinine and GFR between baseline and follow-up presented should be discussed.
As expected, eGFR values decreased as the age of the study sample increased. However, no child/adolescent showed values outside the normal range. It has been shown that creatinine values increase progressively from the first years of life, in accordance with the increase of muscle mass with consequent reduction of eGFR values (Schwartz, George J., George B. Haycock, B. Chir, and Adrian Spitzer. "Plasma Creatinine and Urea Concentration in Children: Normal Values for Age and Sex." The Journal of Pediatrics 88, no. 5 (1976): 828-30. doi:10.1016/S0022-3476(76)81125-0.). Since this is a physiological phenomenon that occurs in all subjects and since BMI has been indexed for age (z-score), we think that, in the absence of renal disease, the physiological reduction of eGFR should not influence our results. In any case, to be sure that this statement was true, we tried adjusting the multiple regression model in Table 4 (Uric acid at follow-up) for eGFR and our finding did not change, despite a significant inverse association between uric acid and eGFR values. See below.
b (95% CI) P
(Intercept) 0.971 0.608 1.335 <0.001
Uric acid at baseline 0.769 0.694 0.843 <0.001
Gender (males) 0.322 0.152 0.491 0.001
Becoming pubescent
during follow-up 0.287 0.094 0.479 0.004
BMI (delta z-scores) -0.420 -0.653 -0.187 <0.001
Delta eGFR -0.009 -0.013 -0.005 <0.001
A sentence has been added to the results section and a paragraph of commentary and one reference have been added to Discussion section.
- Line 353 to 355 - "Similarly, to what observed..." needs clarification.
As requested by the reviewer, this concept has been better clarified, as follows:
“We might have expected that the presence of a diabetic parent would have an impact on the presence and/or persistence of insulin resistance in the offspring. However, familiarity of diabetes mellitus did not influence the results regarding HOMA-index, neither when analyzed as a continuous variable nor as a categorical variable. This finding is similar to what we observed for a family history of dyslipidemia: having a dyslipidemic parent had no significant impact either on the presence of dyslipidemia at baseline or at follow-up.”

Reviewer 3 Report
This is a paper aiming at evaluating the effects of dietary-behavioral treatment on alterations in lipid and glycemic profiles and uric acid values in a population of children and adolescents referred to a second-level outpatient clinic for Cardiovascular Risk Assessment in pediatric age.
Obesity in children and adolescents is gradually becoming a major public health problem in many developing countries. The study of the effects of lifestyle modifications on alterations in lipid and glycemic profiles and uric acid values in a pediatric population at increased cardiovascular risk are highly significant.
This paper follows an appropriate study design. Strengths were related to their anthropometric data that was carefully collected, and their biochemical parameters and caloric intake that were accurately measured. Their capabilities and laboratory facilities to estimate and measure the phenotyping design are paralleled to standards performed in similar labs in other parts of the world. Statistical analysis was appropriately design including multiple linear and logistic regressions.
However, one point with a strong need of clarification is noticed as follows: they collected their data between the baseline and the final follow-up assessment, periodic visits, performed every three months, during which the children’s anthropometric parameters were again recorded. They also disclosed that the study involved 276 subjects consecutively referred from 16/12/2002 to 12/09/2017 (15 years). Separately, they also disclosed that Anthropometric, biochemical, HOMA-Index and uric acid parameters were assessed at baseline and after a median follow-up of 15 months, during which the children received non-pharmacological treatment and attended periodic quarterly control visits. It is not clear how they decided to exclusively present data from these 276 children out of all referrals from 15 years. It appears that they cherry-picked the ones that had the longest follow-up? This is a major weakness from their methodology and study design. They need to strongly clarify this issue in the draft.
They stated that the population of children and adolescents were referred to a second-level outpatient clinic for Cardiovascular Risk Assessment in pediatric age. They continued stating that this Unit included a pediatrician, a cardiologist, a nephrologist and a nutrition expert and that the team interacted to cover all clinical aspects related to the patients. Recent approaches to cardiovascular and diabetes prevention and treatment have shown that beyond biological factors, such as achieved glucose levels, lipids, obesity and blood pressure, it is now necessary to increasingly recognize that sociocultural influences are also important factors in determining an individual’s risk related to the development of cardiometabolic risk. These influences include ethnicity, adopt of attitudes, values, customs or beliefs and behaviors, achieved education level and economic status. This seems to be achieved by the use of specially trained community health workers (CHWs) to promote healthy behaviors that take into account cultural background, psychosocial stressors, and economic limitations that can help overcome many of the barriers for lifestyle change. Perhaps the help of CHWs would have helped achieve a better compliance of the treatment within the three-month interval when the pediatric participants were not seen. The authors need at least to mention this important recent approach in their section related to weaknesses.
A third point is the dietary approach. They stated that a balanced dietary pattern with a precise macronutrients composition (carbohydrates 55%, lipids 30%, proteins 15% of energy intake) and correct content of free sugars, saturated fatty acids and fiber were offered to all participants. Recently, plant-based diets, or better called plant-forward diets, have shown excellent results on their effects on metabolic outcomes. A plant-forward diet has been shown to induce sustained weight loss, effectively prevent and manage metabolic disease, and be well-accepted when compared to a traditional dietary approach. These diets have shown benefits in mitigating hypertension, decreasing plasma lipids, and improving insulin sensitivity and glycemic control. Plant-based eating is associated with lower BMI and adiposity as well. It is advisable to have in the conclusions some paragraph regarding a comment of their nutritional dietary approach with mentions of other avenues such as plant-forward dietary regimes.
Finally, the paper is missing recent research that has increased our awareness of the key role that adipose tissue (AT) dysfunction plays in the development of insulin resistance (IR). The core mechanisms of AT dysfunction involve localized immunometabolic processes during AT expansion, such as impaired angiogenesis and hypoxia, inflammation, inappropriate extracellular matrix (ECM) remodeling, and fibrosis. AT dysfunction is characterized by macrophage infiltration into adipose tissue. Interactions between macrophages and adipocytes represent the early molecular events that will ultimately lead to altered systemic metabolism. Indeed, measurements of adipose tissue (AT) dysfunction phenotypes are far better than measurements of an excess of adipose tissue accumulation as predictors of early CVD and prediabetes. Certain individuals can apparently accumulate an excess of body fat, without developing cardiometabolic disease. They can be considered metabolically healthy despite their high degree of body fat accumulation and their long-standing obesity. This effect can be thought of as healthy AT expansion. On the other hand, unhealthy AT expansion is a major contributor to the systemic metabolic disturbances that are characteristic of obesity and type 2 diabetes. A well-referenced but brief paragraph again in the conclusions would be strongly advisable to show that the authors are aware of the new trends on adipose tissue dysfunction instead of accumulation regarding early detection of cardiometabolic risk.
A very last advice is to recommend the authors to use the term “people with obesity” or “population with obesity” as advised by the latest Guidelines to manage obesity instead of the term “obese” or overweight that may not currently sound politically correct.
Author Response
Reviewer 3
We thank the reviewer for his/her suggestions that allowed us to improve our manuscript.
Comments and Suggestions for Authors
This is a paper aiming at evaluating the effects of dietary-behavioral treatment on alterations in lipid and glycemic profiles and uric acid values in a population of children and adolescents referred to a second-level outpatient clinic for Cardiovascular Risk Assessment in pediatric age.
Obesity in children and adolescents is gradually becoming a major public health problem in many developing countries. The study of the effects of lifestyle modifications on alterations in lipid and glycemic profiles and uric acid values in a pediatric population at increased cardiovascular risk are highly significant.
This paper follows an appropriate study design. Strengths were related to their anthropometric data that was carefully collected, and their biochemical parameters and caloric intake that were accurately measured. Their capabilities and laboratory facilities to estimate and measure the phenotyping design are paralleled to standards performed in similar labs in other parts of the world. Statistical analysis was appropriately design including multiple linear and logistic regressions.
However, one point with a strong need of clarification is noticed as follows: they collected their data between the baseline and the final follow-up assessment, periodic visits, performed every three months, during which the children’s anthropometric parameters were again recorded. They also disclosed that the study involved 276 subjects consecutively referred from 16/12/2002 to 12/09/2017 (15 years). Separately, they also disclosed that Anthropometric, biochemical, HOMA-Index and uric acid parameters were assessed at baseline and after a median follow-up of 15 months, during which the children received non-pharmacological treatment and attended periodic quarterly control visits. It is not clear how they decided to exclusively present data from these 276 children out of all referrals from 15 years. It appears that they cherry-picked the ones that had the longest follow-up? This is a major weakness from their methodology and study design. They need to strongly clarify this issue in the draft.
All children consecutively referred to our center who had two complete assessments (at baseline and follow-up) of all outcomes (anthropometric parameters, complete lipid profile, glycid profile with insulin dosage, and uric acid) were considered for analysis. At the end of follow-up, children who had all parameters in the normal range were discharged, while the others were followed up over time until they reached 18 years of age or achieved all treatment goals. A flow-chart was added as supplementary material to show how many subjects were excluded and for what reason.
They stated that the population of children and adolescents were referred to a second-level outpatient clinic for Cardiovascular Risk Assessment in pediatric age. They continued stating that this Unit included a pediatrician, a cardiologist, a nephrologist and a nutrition expert and that the team interacted to cover all clinical aspects related to the patients. Recent approaches to cardiovascular and diabetes prevention and treatment have shown that beyond biological factors, such as achieved glucose levels, lipids, obesity and blood pressure, it is now necessary to increasingly recognize that sociocultural influences are also important factors in determining an individual’s risk related to the development of cardiometabolic risk. These influences include ethnicity, adopt of attitudes, values, customs or beliefs and behaviors, achieved education level and economic status. This seems to be achieved by the use of specially trained community health workers (CHWs) to promote healthy behaviors that take into account cultural background, psychosocial stressors, and economic limitations that can help overcome many of the barriers for lifestyle change. Perhaps the help of CHWs would have helped achieve a better compliance of the treatment within the three-month interval when the pediatric participants were not seen. The authors need at least to mention this important recent approach in their section related to weaknesses.
We completely agree with the reviewer. Unfortunately, our center is not equipped with this type of support and this was a weakness in our intervention. A sentence on this point has been added to the weaknesses paragraph.
A third point is the dietary approach. They stated that a balanced dietary pattern with a precise macronutrients composition (carbohydrates 55%, lipids 30%, proteins 15% of energy intake) and correct content of free sugars, saturated fatty acids and fiber were offered to all participants. Recently, plant-based diets, or better called plant-forward diets, have shown excellent results on their effects on metabolic outcomes. A plant-forward diet has been shown to induce sustained weight loss, effectively prevent and manage metabolic disease, and be well-accepted when compared to a traditional dietary approach. These diets have shown benefits in mitigating hypertension, decreasing plasma lipids, and improving insulin sensitivity and glycemic control. Plant-based eating is associated with lower BMI and adiposity as well. It is advisable to have in the conclusions some paragraph regarding a comment of their nutritional dietary approach with mentions of other avenues such as plant-forward dietary regimes.
So-called "plant-diets" have been shown to have a protective effect on the risk of type 2 diabetes, cardiovascular disease, and obesity in adults (J. Harland and L. Garton An update of the evidence relating to plant-based diets and cardiovascular disease, type 2 diabetes and overweight 2016 British Nutrition Foundation Nutrition Bulletin, 41, 323–338). However, the authors themselves acknowledge the difficulty in defining exactly what a plant-diet is. In fact, this term, beyond its ethical and ecological values, includes very different food choices: from vegetarian and vegan diets to Mediterranean Diet and DASH diet. A vegan diet can potentially be critical in children because of inadequate intake of protein, long-chain fatty acids, iron, zinc, vitamin D, iodine, calcium and in particular vitamin B12. (T A Sanders. Growth and development of British vegan children. Am J Clin Nutr. 1988 Sep;48(3 Suppl):822-5. doi: 10.1093/ajcn/48.3.82), although some scientific societies allow it, but only if properly integrated and controlled (M Amit; Canadian Paediatric Society, Community Paediatrics Committee. Vegetarian diets in children and adolescents. Paediatr Child Health 2010;15(5):303-314). Vegetarian diets in adults are assumed to be healthy; however, it is unclear whether this is also true for children and adolescents. Existing data on this topic do not allow to reach a solid conclusion on the health benefits or risks of vegetarian diets on the nutritional and health status of children and adolescents (S Schuermann, M Kersting, U Alexy. Vegetarian diets in children: a systematic review. Eur J Nutr. 2017 Aug;56(5):1797-1817). In our study we chose to propose a Mediterranean Diet, as it is closer to the culture of our country, including specific indications based on metabolic alterations present in individual cases. However, we have always prescribed an increase in the consumption of fruits and vegetables.A few paragraphs regarding this point have been added to the discussion.
Finally, the paper is missing recent research that has increased our awareness of the key role that adipose tissue (AT) dysfunction plays in the development of insulin resistance (IR). The core mechanisms of AT dysfunction involve localized immunometabolic processes during AT expansion, such as impaired angiogenesis and hypoxia, inflammation, inappropriate extracellular matrix (ECM) remodeling, and fibrosis. AT dysfunction is characterized by macrophage infiltration into adipose tissue. Interactions between macrophages and adipocytes represent the early molecular events that will ultimately lead to altered systemic metabolism. Indeed, measurements of adipose tissue (AT) dysfunction phenotypes are far better than measurements of an excess of adipose tissue accumulation as predictors of early CVD and prediabetes. Certain individuals can apparently accumulate an excess of body fat, without developing cardiometabolic disease. They can be considered metabolically healthy despite their high degree of body fat accumulation and their long-standing obesity. This effect can be thought of as healthy AT expansion. On the other hand, unhealthy AT expansion is a major contributor to the systemic metabolic disturbances that are characteristic of obesity and type 2 diabetes. A well-referenced but brief paragraph again in the conclusions would be strongly advisable to show that the authors are aware of the new trends on adipose tissue dysfunction instead of accumulation regarding early detection of cardiometabolic risk.
A short paragraph and some references were added to the discussion to develop this point, as requested by the reviewer. However, we think that the concept of "metabolically healthy obesity" should be considered with caution. In fact, it has been demonstrated that non-negligible percentage of “metabolically healthy” children with obesity has high values of non-traditional risk factors such as HOMA index and serum uric acid. In addition, it is possible that, in the absence of appropriate intervention and with the elapsing of time (do not forget that we are dealing with very young subjects) a "metabolically healthy" individual turns into a "metabolically unhealthy" individual.
A very last advice is to recommend the authors to use the term “people with obesity” or “population with obesity” as advised by the latest Guidelines to manage obesity instead of the term “obese” or overweight that may not currently sound politically correct.
We are in agreement with the reviewer. Terminology has been corrected as requested.
